# Analysis of factors affecting visual comfort in hotel lobby

**Ziwen Geng**[1]*, **Wei Le**[2]*, **Benhai Guo**[2], **Hongjuan Yin**[2]

**1** College of Air Transport and Engineering, Jincheng College of Nanjing University of Aeronautics and Astronautics, Nanjing, China, **2** College of Economics and Management, China Jiliang University, Hangzhou, China

* 1529427377@qq.com (ZG); lewei@cjlu.edu.cn (WL)

**Data Availability Statement:** All relevant data are within the manuscript and its Supporting Information files.

**Funding:** The authors received no specific funding for this work.

## Abstract

From the perspective of emotion, utilizing eye tracking technology, this paper proposes 12 different 3D hotel lobby models to investigate how would the light illuminance, wall color, decoration style and music genre affect the visual comfort specifically. The experiment results show that the illuminance of the lamp, the color of the wall, and the decoration style have a significant impact on visual comfort. The music genre would not affect consumer's visual comfort perception of lamp illuminance, wall color and decoration style. Normal illuminance brings the most comfort, followed by bright illuminance, and then dim illuminance; Warm color brings more comfort than cool color; and Chinese style lobby brings more comfort than European style.

## Introduction

Hotel lobby visual comfort is an important factor to determine the first impression of consumer experience, which not only affects consumers' initial purchase decision, but also relates to the formation of consumers' patronage motivation. It has been one of the hot topics in academic circles. After summarizing the factors influencing hotel's comfort, we found that visual conformation affects consumer perception and plays the most important role in sensory evaluation factors. However, visual comfort, which ultimately affects people's motivation to consume and how visual setting affects visual comfort, was often overlooked. Although there have been lot of visual setting studies on restaurants, resorts and hotels, Durna et al.(2015) believed that more empirical research was needed to investigate the impacts of different servicescape elements on customer response [1]. Especially for the service industry of hotel, the main stimuli of the hotel lobby includes: color, illuminance, decoration style and spatial scale [2]. In the design of a hotel, decoration style would be determined firstly; In restaurants and the other similar scenes, lamp illuminance and color would be the key visual factors affecting customer sentiment and behavioral intentions, and also the important factors of creating a warm atmosphere. There is an interaction between consumers' senses [3, 4]. For instance, Music may affect the visual comfort brought by illuminance and color. Compared with altering size of hotel lobby space, changing music genre, illuminance and wall color could improve consumers' comfort more cost-effectively and easily. Generally speaking, previous studies on comfort

**Competing interests:** The authors declare no conflict of interest.

in hotel industry only involved one factor such as color or lighting. However, these factors have been proved to exert significant impact on consumer sentiment and satisfaction, but in reality, we experience the overall comfort of hotel lobby, not only the comfort brought by illuminance or color. Therefore, we study the illuminance level, wall color, decoration style and music genre in one model.

After understanding the factors that influence visual comfort, we also need to address how to measure comfort scientifically. Previous studies, which had some drawbacks, measured comfort only by subjective questionnaires. On the one hand, it is difficult for consumers to clearly remember the comfort feelings at the time. On the other hand, their responses to certain comfort are unconscious and are difficult to distinguish, and lack timeliness and effectiveness. Therefore, the scientific and rational measuring method is important in enhancing of visual comfort research. Visual comfort can be defined as "subjective conditions of visual happiness caused by the visual environment" [5]. A lot of psychological evidences show that visual comfort and emotion are inextricably linked. For example, in a visually comfortable environment, people will feel relaxed, happy and excited [6]. Therefore, we take advantage of eye tracking technology to measure comfort mainly for two reasons. Firstly, there have been considerable mature studies proving that eye movement indicators can measure emotion, which could imply comfort. Secondly, Vytautas Abromavičius and Artūras Serackis (2018) showed that pupil diameter and the number of fixation points could be used to measure visual comfort [7]. In conclusion, based on the relationship between visual comfort and emotion, this study utilizes eye-tracking technology (pupil diameter and fixation points) and questionnaires to measure the effects of light intensity, warm and cool color, decoration style and music style on visual comfort.

## Research on the influencing factors of hotel visual comfort

As customers are motivated by everything for good feelings and comfort, hospitality businesses need to invest heavily in space design according to their marketing strategies. Specifically, as for the aspect of sensory, hotel operators can choose illuminance, color, fixed decorations, and the other factors affecting the overall comfort experience of customers [8]. Comfort has been cited widely in the hospitality industry, but the studies on the impact of hotel visual comfort is very rare. The interpretation of comfort is also narrow [9]. Comfort is a state in which people and environment are relatively balanced in psychology, physiology and physics. When human body is in balance, the feeling could be called comfort. If this balance is broken by external factors, an uncomfortable feeling would arise [6]. In this paper, visual comfort is defined as the physiological and psychological pleasure balance caused by external visual stimuli such as illuminance and color.

Vision often provides the primary sensory cue of lobby. Relevant statistics show that more than 80% of the information acquired by the human body comes from vision, and vision is the most important perception of humans and animals. Diţoiu and Căruntu (2014) interviewed 500 young people and found that when they selected destinations, visual elements accounted for 98.2% of the perceptual evaluation factors [2].

In the study of hotel comfort impact, visual comfort is the most crucial sensory dimension, but visual comfort is often neglected and rarely studied. In residential environments, the visual comfort of lighting is most affected by illuminance and color temperature [10]. Under high illuminance, which is more conducive to the relief of fatigue, and the level of anxiety would decline. Providing a user-friendly and user-controlled lighting system for hotel rooms helps to improve customer satisfaction [11]. Siamionava et al. (2018) found that hotel wall color would affect customers' perception. Participants prefer to stay in the blue room [12]. Li et al.(2015)

suggested that the decorative style would affect consumers' purchase intention [13]. The warm atmosphere created by lighting, color and other factors could guide customers' behavior. Through the analysis of the key visual dimensions of the hotel lobby, it is found that the light illuminance, wall color and decoration style are the top three factors affecting customer comfort. There is an interaction between the senses. Humans are exposed to multi-sensor stimuli in the environment affecting their perception. Vision would affect the sense of taste, dim lighting would enhance the sense of taste, and the perception of auditory quality affects the perception of visual quality [14]. For example, music would affect customers' perception of light and color [15, 16].

In recent years, experiential marketing in the hospitality industry has developed rapidly [17, 18]. When choosing a hotel, customers consider not only cognitive attributes (e.g., price, food quality, services, and national brand), but also affective (e.g., comfort and entertainment) and sensory attributes (e.g., overall atmosphere, room quality) [19]. Moreover, the servicescape is an effective prerequisite for formation of a good impression and pleasure of the customer [20]. Alfakhri et al.(2018) explored the lived experience of design and art in the hotel landscape. The results showed that the interior design elements would trigger consumers' emotions (i.e., entertainment, relaxation and satisfaction), which in turn would affect their behaviors (i.e., loyalty, time spent, price sensitivity, social interactions, and word of mouth) [21]. Therefore, it is important to understand how ambience affects the consumers' experience about comfort. This paper is dedicated to exploring the influences of different hotel lobby designs. Specifically, we will focus on three different lighting illuminance (bright, normal, dim), two wall colors (cool, warm) and two styles of decoration (Chinese style and European style) in the environment of Chinese or European style music.

## Physiological measures of visual comfort and mood

Emotion is conceptualized as a multi-component response to an emotionally potent antecedent event, causing changes in subjective feeling quality, physiological activation and expressive behavior [22]. In order to better clarify the mechanism of emotion and motivation, Izard (2013) proposed "Different Emotions Theory (DET)", arguing that emotions comprised of facial expressions, brain and nerve related activities and emotional experience. He proposed the "emotion-cognitive-motor response" model. He believed that emotion was the basic motivation and cognition played an important role in process. The interaction of emotion, cognition and movement system produce a certain of experience, emotion and response [23]. Customer emotional experience is generated through the interaction of cognition, sensory choices and affection in the hotel context. It is important to examine the consumers' choice behavior through a comprehensive understanding of the interaction of these different attributes [19]. In order to better explain the relationship between service scenarios, emotional responses and behavioral intentions, Mehrabian and Russell (1974) proposed a Stimulus-Organism-Response (S-O-R) theory model. In the SOR model, the relationship between stimulation, body and response can be interpreted as three phases: the body's perception of environmental stimuli, the interpretation of visual information as emotions, and the response to stimuli based on emotional responses [24]. As an important part of the visual service scene, Knez and Hygge (2002) showed that indoor lighting could affect emotional, cognitive processes and physiological functions. Specific color triggers specific emotions, and red (as opposed to blue) triggers more active physiological changes [25]. Warm colors (especially red) can trigger excitement, while cool colors (especially blue) are associated with feelings of relaxation, peace and pleasure [26]. Music also changes physiological indicators such as human heart rat [27].

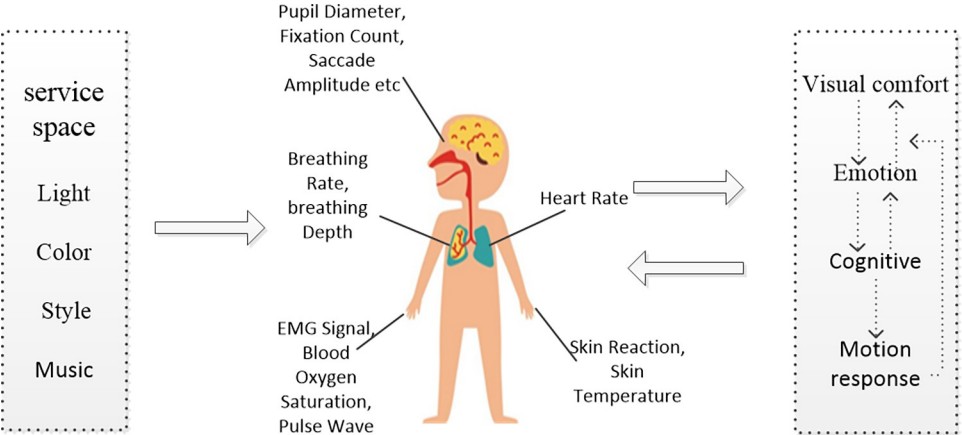

**Fig 1. Physiological mechanism of stimulation-emotion-perception.** (Picture from web search (https://detail. youzan.com/show/goods?alias=2fmniq6pospdh&activity=)).

Emotional changes are mainly controlled by the sympathetic nervous system, which in turn affects changes in eye movement indicators. Therefore, it is a reasonable and effective way to observe emotional changes through eye movement indicators. Traditional emotion measurement mainly adopts self-reporting method. With the maturity of emotion theory and technology, the requirements for the accuracy of user emotion data measurement have been increasing gradually. Various objective physiological measurement methods, such as eye tracking, multi-channel physiological instruments, EEG and other physiological index methods have been used to monitor emotion-evoked physiological signals to measure user emotions [28]. Research by Vytautas Abromavičius and Artūras Serackis (2018) has shown that pupil diameter and the number of fixation points could be used to measure visual comfort. The "uncomfortable" visual comfort score corresponds to the largest pupil size and less fixation points [7]. For the visual aspect, the favorite brand could bring more pleasant visual perception than ordinary brand, and the associated blinking amplitude is tremendously reduced [28]. Compared with subjects with negative emotions, subjects with positive emotions observe the target for a longer time and a wider range [29]. There is a linear relationship between customer's pupil size and aesthetic pleasure. When the image is assessed as pleasant and comfortable, the pupil tends to expand, otherwise the pupil tends to narrow [30].

In the service and leisure environment, emotion is one of the core elements of customer satisfaction [31]. Hotel guests' sentiment constitutes an important part of satisfaction and loyalty during their stay [32]. Emotion is a key factor affecting the success of the service industry. When individual's mood and comfort change, the corresponding physiological indicators will also change, such as pupil diameter (the pupil diameter is used to describe the size of the pupil) and number of fixation points (the fixation point is the stop point of the eye on the observation target). Usually, when the duration of the stay points is more than 100 ms, it is called fixation (the number of fixation points is the number of these stay points) [33]. Through the specific changes of these indicators, individual emotional changes can be measured, which provides a theoretical basis for the study (see Fig 1).

### Influence of light illuminance, wall color, decoration style and music genre on mood and comfort

Lighting, which affects the overall impression and perceived comfort of the lobby, is an indispensable factor in hotel lobby design [11, 34], and lighting is the most effective factor in

motivating customers to consume. Ambient light affects visual input and may vary in color, distribution, and glow. Compared with the other factors, lighting comfort is affected by illuminance and color temperature more [10]. Low-illuminance light is more likely to cause visual fatigue. It is easier for color of lower-contrast with darker lighting and simple decoration to create a relaxed and romantic atmosphere [4]. Some scholars have also found that warmer and brighter light bring a stronger sense of well-being and pleasure, and bright light can make subjects less sleepy and more energetic [3]. For color temperature, Yu and Akita (2019) argued that higher spatial lighting correlated with higher color temperature. A lower color temperature produces a higher sense of security, positivity, and tranquility [11]. Light intensity affects people's feeling and emotion physiologically and psychologically [35]. Specifically, compared to room with cool white light (relatively high color temperature), room with warm white light tends to be perceived as more positive (i.e. pleasant, attractive, and relaxing) atmospheres [36].

In addition to good visibility and visual comfort, as well as creating a pleasant ambience and an aesthetically pleasing environment, good ambient lighting should also stimulate emotion [37]. In addition to affecting consumers's sleep and mood indirectly, light can regulate mood by activating brain regions involved in emotional processing directly (i.e., the medial amygdala and lateral habenula) [36].

The interaction of light and sound also affects people's perception on environment. Different environments imply different lighting requirements. In real life, some hotel lobby lighting design is too bright and dazzling, some are too dim, imposing people an uncomfortable feeling. Light illuminance affects the mood and visual comfort of consumers. Based on this, this paper proposes the following assumption:

H1. The illuminance level of hotel lobby would affect the visual comfort of the consumer.

Color is one of the most influential factors in object and space recognition, and shaping understanding by color contrast is often more effective than by illuminance contrast. color in man-made environments could play various roles, such as identification, symbolism, semiotics, emotional control, physical and mental comfort, and communication [37]. Color has a strong impact on consumers' visual comfort, emotion, value perception, and behavior. The emotional effect of color is particularly important in hotel industry, as emotion is very important for customers' satisfaction [12]. Among them, color has the most significant impact on the perception of hotel lobby atmosphere [34]. In luxury hotel aesthetics, cool color is better and more flattering than warm color. In addition, warm color is cheaper than cool color in consumers' minds, so they usually prefer warm color to cool color [38]. Through a study of 496 participants, Tantanatewin and Inkarojrit (2018) found that warm-colored restaurant scenes would give customers a higher level of pleasure, and the pleasure brought by the color would increase customers' choices of entering the restaurant, and it would be easier to satisfy consumer [39]. There was an interaction between color and multiple factors, for example, the interaction between light and color revealed that warm color under bright illuminance can cause negative perceptions and reactions from customers, and users' evaluation, excitement and willingness to purchase are reduced correspondingly, but the warm color under soft lighting produces diametrically opposite effects [40]. The interaction between music and color showed that participants felt more excited and pleasant in fast music and warm color conditions than those in slow music and cool color environments. Furthermore, the consistency of these two factors enhanced the effect of atmosphere on people's emotional responses. However, for hotel lobbies, the visual factor is more important, and therefore the interaction between lighting and color is more meaningful [41].

Consumer perception of color creates a mental process that produces positive behavior. This process and behavior happen instinctively [42]. Colors has a significant impact on mood and visual comfort, and customers prefer warm color. Therefore, in real life, whether in hotel

or restaurant, warm color is often used to create a warm atmosphere, but the warm color under bright illuminance would reduce customers' comfort. Therefore, it is speculated that under the non-bright illuminance, the warmth of the hotel lobby is more visually comfortable. Based on this, the following two assumptions are made:

H2a. The visual comfort of a warm-colored hotel lobby is greater than that of a cool-colored one.

H2b. In the normal light atmosphere, the warm-colored hotel lobby is more comfortable than the cool-colored one.

Ambient atmosphere and interior design are defined as hotel landscape. Growing interest in service landscapes has prompted luxury hotels to enhance the uniqueness of their rooms, and invest heavily in the design and interior of lobbies and public areas to meet the diversity of customers' aesthetic values [38]. Countryman and Jang (2006) studied the atmosphere of hotels and accommodations using scenes and photographs of hotel lobbies, identifying the elements in the hotel lobbies that contributed the most to the overall impression. Of the six elements they examined, layout, style, color, lighting and furniture, three were found to be unique and important at a level of 5% or more—style, color, and lighting [34]. Therefore, each hotel lobby used light, color, sound, spatial layout and function and signs to the extreme, creating a unique emotional environment, impressing consumers through sensory stimulation, and attracting consumers to visit again. The decoration style of the hotel would affect the consumers' willingness to stay. Especially in the cultural atmosphere of China, compared with the higher conspicuous decorative style, the lower conspicuous decorative style can stimulate customers' willingness to purchase hotel's service [13].

Style is a key element of interior design influenced by a variety of factors. Because individuals are very different in style preferences, some prefer Chinese style, and others prefer European style. Different styles of lobby with different genres of music would have very different effects. If the hotel style caters to consumers' aesthetics, it would create a comfortable and pleasant atmosphere to bring happiness and peace for customer. Based on this, the following assumption is made:

H3. The decoration style of hotel lobby would affect the visual comfort of consumers.

The company tries to create unique emotional environments in which the music matches the architecture, lighting, color and corporate identity. For example, many hotels and restaurants employ professional sound designers to create unique and personalized musical environments. Famous hotels and restaurants around the world have adopted perceptual strategies to select music matching the entire atmosphere to enhance the customer's experience of services. Interactions between the senses are mainly manifested in their direct influence on mood [43]. While lighting, color and decoration style are all the key visual factors, and they form the overall sensory stimulation that hotel lobbies intend to bring to guests. Therefore, it is not enough to study only one factor. For example, music often interacts with other environmental factors, and musical emotion is significantly associated with lighting [16, 44]. There are also structural correspondences (such as emotional expression, hierarchical organization, and contrast) between color and music [45]. Using a combination of music and color to maximize their emotional impact. Emotion will be enhanced if the lighting corresponds to the music enjoyed by audience, as the combination of music and lighting promotes the emotional impact [15]. Demoulin (2011) studied the consistency of music and the overall atmosphere and showed that consistency between music and the atmosphere of the service landscape was crucial. Higher musical consistency led to lower arousal and greater happiness. Music that aligns with the atmosphere of servicescape creates a feeling of relaxation, calm and joy. Happy customers rate the service environment and service quality higher. Musical consistency improves directly customer's perception on service quality [43].

In summary, music tends to interact with visual factors such as lighting, color, and style in the environment. The same visual environment with different music may have a very different effect. When the music genre is consistent with the decoration style of the hotel lobby, it creates a pleasant and comfortable atmosphere. Therefore, this paper proposes the following assumptions:

H4a. Compared to the Chinese-style hotel lobby where European music is played, the Chinese-style hotel lobby where Chinese music is played brings more visual comfort.

H4b. Compared to the European-style hotel lobby where Chinese music is played, the European-style hotel lobby where European music is played brings more visual comfort.

H4c. Compared to the European-style hotel lobby where European music is played, the Chinese-style hotel lobby where Chinese music is played brings more visual comfort.

## Methods

### Participants

The total number of valid participants was 43 and they all had hotel experience, and they were paid volunteers, including 28 young-adult Chinese female university students (Mean age = 22.54 years, SD = 2.24) and 15 young-adult Chinese male university students (Mean age = 22.80 years, SD = 3.19). The participants had no obvious preference for Chinese-style or European-style hotels. All participants had normal or corrected-to-normal vision.

### Ethics statement

This experiment was approved by the Review Board of China Jiliang University, the participants informed consent in written. Our study adhered to the Declaration of Helsinki.

### Experimental design

This experiment was 3(light illuminance: bright, normal, dim) × 2(wall color: cool, warm) × 2 (decoration style: European, Chinese) × 3(music: no music, Chinese music, European music), a four-factor within-group design experiment. The experiment employed eye-tracking technology, and collected data by questionnaires. The purpose of the experiment was to evaluate lighting illuminance (bright, normal, dim), wall color (cool, warm), decoration style (European style, Chinese style) and the influence of the two styles of music on visual comfort. This paper draws on the PANAS (The Positive and Negative Affect Scale) emotional self-rating scale (Watson et al, 1988) to measure emotional response [46], where the dependent variables were visual comfort and two eye movement indicators (pupil diameter, number of fixation points).

### Apparatus and stimuli

The Instrument used was an Germany-Made IViewX hi-speed Eye Tracking System made by SensoMotoric Instrument (SMI). The technology basis of the eye-movement tracking system was infrared corneal and pupil reflex technology. The sampling frequency was 500Hz. The visual stimuli were presented on a 17-inch Cathode ray tube display with a resolution of 1280 × 1024 pixels. In the experiment, one notebook screen was employed to display stimuli and record data, one screen was used to display stimuli, and the other screen was used to record eye movement data. Experiments were designed and played by the Experiment Center software, namely, iViewX3.5 software, for eye calibration and recording of experimental eye movement data, and BeGaze3.5 software was used for extraction and analysis of eye movement data.

**Table 1. Color indicators of room models.**

| Serial Number | Variable | Hue | Saturation | Value | RGB | Color Number |
|---|---|---|---|---|---|---|
| 1 | Cool color | 26˚ | 0 | 0 | R: 255 | #ffffff |
| | | | | | G: 255 | |
| | | | | | B: 255 | |
| 2 | Warm color | 26˚ | 34 | 19 | R: 223 | #dfb594 |
| | | | | | G: 181 | |
| | | | | | B: 148 | |

Note. R is the intensity value of the red channel. G is the intensity value of the green channel. B is the intensity value of the blue channel

A 3D hotel lobby model was based on a compilation of typical hotel lobby, and 12 3D lobby models were made by 3d Max software according to three variables: 3(light intensity: bright, normal, dim) × 2(wall color: cool color, warm color) × 2(decoration style: European, Chinese). The size of models was 2400 × 1500 in pixel. Firstly, in order to design a more reasonable hotel lobby model, we invited a hotel design expert to design the first draft of the model. Secondly, we invited participants to evaluate whether the lighting illuminance, wall color and decoration style of the hotel lobby model met the cognitive standards, and whether the hotel lobby design was reasonable. According to the opinions of the participants, we adjusted the hotel lobby model and added a serviceman behind the counter, reduced the light intensity of normal lighting and dim lighting in the lobby.

In order to verify the scientific validity of the experiment and the questionnaire, the researcher conducted a pre-experiment before the formal experiment. The pre-experiment intended to achieve the following three purposes: 1. The subjects evaluating whether the experimental process and time setting were reasonable. 2. The subjects assessing whether the description of the questionnaire was accurate and clear. 3. offering subjects 6 Chinese-style and European-style songs, and asking them to choose two more suitable songs.

Finally, the color value of cool color room is 0 and that of warm color room is 19. The specific color index is shown in Table 1. Light intensity is the three indicators determined according to the feelings of the subjects in the preliminary experiment. The bright light intensity is 13000cd, the normal light intensity is 9000cd, and the dim light intensity is 2000cd. Chinese music is "ErhuChant", European music is Chopin's "Nocturne".

## Procedures

The experiment was performed in an eye movement laboratory with constant temperature and illuminance. All stimulation materials were displayed on the computer screen through the Experiment Center software. The experimental process includes five groups of sub-experiments (1. Observing the 12 hotel lobby models without music. 2. Observing the 6 Chinese-style lobby models in which Chinese music was played. 3. Observing the 6 Chinese-style lobby models in which European music was played. 4. Observing the 6 European-style lobby models in which Chinese music was played. 5. Observing the 6 European-style lobby models in which European music was played.). Each group of experimental data collection consisted of two parts.

Before the experiment, Part 1, the subjects familiarized themselves with the laboratory environment and experimental requirements firstly and then subjects sat in a comfortable chair. Setting the distance between the eyes and the screen to 60 cm. When the five-point gaze tracking correction reached the standard level, the formal experiment started. During the formal experiment, participants would see detailed experimental instructions on the screen, and the presentation time is 5s ("You are a consumer who is preparing to stay at the hotel. Then you

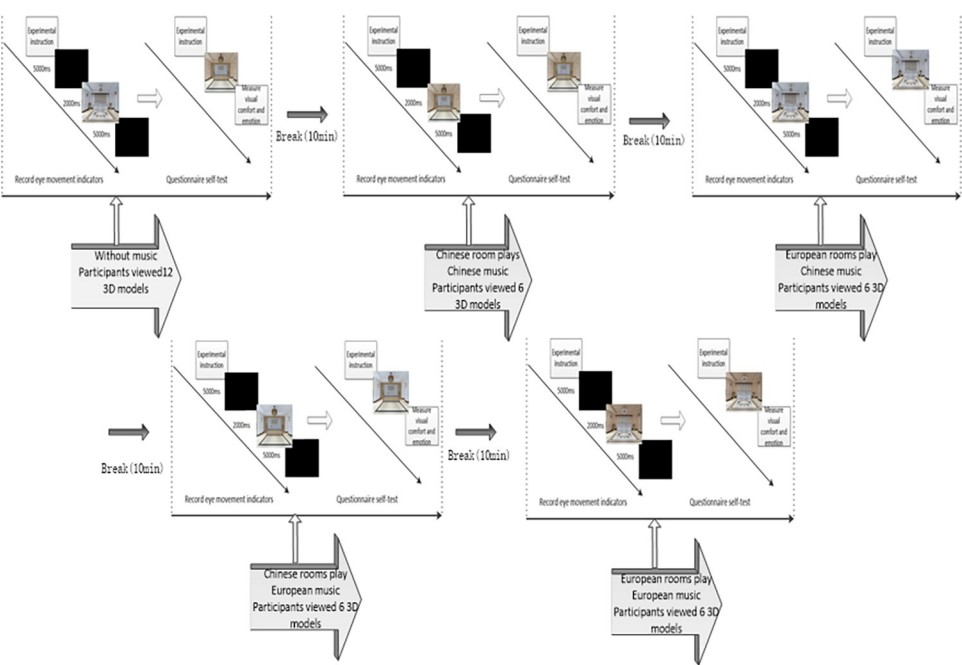

**Fig 2. Flow chart of experimental stimulus presentation.**

will walk into the hotel lobby and see the lobby environment. The lobby model will be automatically presented after 7s."), a black screen for 2 seconds appeared, followed by stimulus images (the presentation time of each stimulus image was 5s, and a 2s black screen appeared between alternate images). The second part, the stimulus picture was presented again, and the participants verbally explained their emotions and comfort experience to the experimenters. The sentiment scale included eight emotional questions based on the PANAS emotional self-rating scale (Watson et al., 1988) [46], which were modified according to research needs, and the Likert 7-point scale for arousal. After offering the answer, the subjects rested 10 minutes for the next set of experiments (see Fig 2). After the experiment, the data was imported into R3.5.0 for analysis of variance.

## Eye movement recording

Before the formal experiment, the position and angle of the camera and the screen should be adjusted so that the binocular image was centered on the screen and stable. A slight movement of the subject's head would not cause the projection to be lost, and the subject's blink could be recovered quickly. A nine-point calibration and validation procedures were performed before starting each block so as to ensure the reliability of the eye tracker and data accuracy. When the X and Y direction calibration accuracy was less than 0.5 degrees, it reached the best status; if the X and Y direction errors were within 1.0 degree, it was qualified; if one direction error was greater than 1.0 degree, it needed to be recalibrated.

## Results

### Eye movement data analysis

The total effective test subjects were 43 people. Subjects' eye movement data was imported into R3.5.0 and SPSS25.0 for ANOVA, with Bonferroni correction applied for post hoc tests.

**Table 2. ANOVA results for pupil diameter and number of gaze points under different types of stimulation.**

| Variables | Df | MS | F | p |
|---|---|---|---|---|
| 2×3×2×2 ANOVA results for pupil diameter | | | | |
| Light illuminance | 2 | 91.80 | 263.56 | < .0001*** |
| **Variables** | **Difference** | **SE** | **df** | **p** |
| Post-event comparison test of light intensity | | | | |
| Bright illuminance–Dim illuminance | -1.03 | 0.03 | 84 | < .0001 |
| Bright illuminance–Normal illuminance | -0.42 | 0.03 | 84 | < .0001 |
| Dim illuminance–Normal illuminance | 0.61 | 0.03 | 84 | < .0001 |
| **Variables** | **Df** | **MS** | **F** | **p** |
| Wall color | 1 | 0.95 | 2.74 | 0.10 |
| Decoration style | 1 | 0.28 | 0.81 | .369 |
| Music genre | 1 | 0.44 | 1.27 | .260 |
| 2×3×2×2 ANOVA results for the number of gaze points | | | | |
| Light illuminance | 2 | 20.61 | 62.87 | < .0001*** |
| **Variables** | **Difference** | **SE** | **df** | **p** |
| Post-event comparison test of light intensity | | | | |
| Bright illuminance–Dim illuminance | 3.77 | 0.58 | 84 | < .0001 |
| Bright illuminance–Normal illuminance | -0.11 | 0.58 | 84 | 0.9804 |
| Dim illuminance–Normal illuminance | -3.88 | 0.58 | 84 | < .0001 |
| **Variables** | **Df** | **MS** | **F** | **p** |
| Wall color | 1 | 2.36 | 7.20 | 0.007* |
| **Variables** | **Difference** | **SE** | **df** | **p** |
| Post-event comparison test of wall color | | | | |
| Cool color—Warm color | 1.84 | 0.40 | 42 | < .0001 |
| **Variables** | **Df** | **MS** | **F** | **p** |
| Decoration style | 1 | 0.08 | 0.23 | .629 |
| Music genre | 1 | 0.25 | 0.77 | .381 |

Note.

***p< 0.001

**p< 0.01

*p< 0.05.df is the degrees of freedom;MSE is mean square error; The F value is the statistic for the F test; Difference is the difference in the post hoc comparison test; SE is Standard Error.

Analysis of the results of direct pupil and number of gaze points showed that the main effect of the light illuminance was significant at the 0.01% level, and the main effect of wall color was significant at the 5% level, but the main effect of decorative style was not significant. According to the results of post-mortem analysis, the number of gaze points was lowest in the dim room ($\bar{x}_{\text{dim illuminance}}$ = 9.83) and slightly higher in the normal room ($\bar{x}_{\text{normal illuminance}}$ = 13.70) than in the bright illumination room ($\bar{x}_{\text{bright illuminance}}$ = 13.59). For wall color, subjects had fewer gaze points ($\bar{x}_{\text{warm color}}$ = 11.456) in warm-toned rooms.(see Table 2).

The interaction among lighting illuminance, wall color and decoration style was significant. Bonferroni-corrected post hoc analysis results showed that in the lobby of Chinese style, bright illumination cool color ($\bar{x}_{\text{Chinese decoration style–bright illumination–cool color}}$ = 18.76) caused the highest number of viewpoints, followed by normal illumination cool color ($\bar{x}_{\text{Chinese decoration style–normal illumination–cool color}}$ = 15.64), and dim illumination ($\bar{x}_{\text{Chinese decoration style–dim illumination–warm color}}$ = 9.38) remained the lowest. In the European-style hotel lobby, normal illumination warm color ($\bar{x}_{\text{European decoration style–normal illumination–warm color}}$ = 15.59)

**Table 3. Interaction effects and post hoc ANOVA results.**

| Variables | Df | MS | F | p |
|---|---|---|---|---|
| 2×3×2×2 ANOVA results for pupil diameter | | | | |
| Light illuminance: Wall color | 2 | 0.14 | 0.41 | .664 |
| Wall color: Music genre | 1 | 0.02 | 0.06 | .800 |
| Light illuminance: Music genre | 2 | 0.07 | 0.21 | .808 |
| Decoration style: Music genre | 1 | 0.36 | 1.02 | .312 |
| Light illuminance: Wall color: Decoration style | 2 | 1.94 | 5.58 | .004* |
| Light illuminance: Wall color: Music genre | 2 | 0.05 | 0.13 | .879 |
| Light illuminance: Decoration style: Music genre | 2 | 0.11 | 0.30 | .730 |
| Light illuminance: Wall color: Decoration style music genre | 2 | 0.05 | 0.14 | .868 |
| 2×3×2×2 ANOVA results for the number of gaze points | | | | |
| Light illuminance: Wall color | 2 | 0.09 | 0.28 | .756 |
| Light illuminance: Decoration style | 2 | 2.75 | 8.39 | < .0001*** |
| Wall color: Decoration style | 1 | 0.78 | 2.38 | .123 |
| Light illuminance: Music genre | 2 | 0.14 | 0.43 | .648 |
| Wall color: Music genre | 1 | 0.10 | 0.31 | .574 |
| Decoration style: Music genre | 1 | 0.18 | 0.55 | .46 |
| Light illuminance: Wall color: Decoration style | 2 | 11.96 | 36.47 | < .0001*** |
| Light illuminance: Wall color: Music genre | 2 | 0.57 | 1.73 | .178 |
| Light illuminance: Decoration style: Music genre | 2 | 0.01 | 0.04 | .960 |
| Light illuminance: Wall color: Decoration style music genre | 2 | 0.05 | 0.14 | .868 |

Note.

***p< 0.001

**p< 0.01

*p< 0.05.df is the degrees of freedom;MSE is mean square error; The F value is the statistic for the F test; Difference is the difference in the post hoc comparison test; SE is Standard Error.

led to the highest number of gaze points, and dim illumination warm color ($\bar{x}_{\text{European decoration style−dim illumination−warm color}}$ = 8.14) were the lowest. In contrast to the predicted findings, there was no interactive effect between decoration style and music genre (see Tables 3 and 4).

## Analysis of questionnaire data

Conducting repetitive analysis of variance within the test for comfort, the results showed that the main effect of the light illuminance, wall color and decoration style were significant at the 0.01% level, but music genre had no significant effect on visual comfort. Consistent with the post hoc analysis of eye-movement data, dim lighting ($\bar{x}_{\text{dim illuminance}}$ = 1.756) was the least comfortable, and there was no significant difference between bright ($\bar{x}_{\text{bright illuminance}}$ = 4.509) and normal illumination ($\bar{x}_{\text{norma illuminance}}$ = 4.564). The warm color ($\bar{x}_{\text{warm color}}$ = 3.756) or Chinese lobby ($\bar{x}_{\text{Chinese style}}$ = 3.775) was more comfortable and the music did not affect the subjects' comfort level (see Table 5).

For the interaction, the interaction among lighting illuminance, wall color and decoration style was significant, and there was a pairwise interaction effect. Bonferroni-corrected post hoc analysis results showed that under normal light illumination, a warm ($\bar{x}_{\text{normal illuminance−warm color}}$ = 4.680) or Chinese style ($\bar{x}_{\text{normal illuminance−Chinese style}}$ = 4.657) hotel lobby would bring higher comfort level. Under bright light, warm colors ($\bar{x}_{\text{bright illuminance−warm color}}$ = 4.895) continued to bring

**Table 4. Interaction and post-hoc ANOVA results of lighting illuminance and decoration style.**

| Variables | Df | MS | F | p |
|---|---|---|---|---|
| 2×3×2×2 ANOVA results for gaze points | | | | |
| Light illuminance: Decoration style | 2 | 2.75 | 8.39 | < .0001*** |
| **Variables** | **Difference** | **SE** | **df** | **p** |
| Post-mortem contrast test of lighting intensity and decoration style | | | | |
| Chinese decoration style | | | | |
| Bright illuminance–Dark illuminance | 7.00 | 0.75 | 160.69 | < .0001 |
| Bright illuminance–Normal illuminance | 1.96 | 0.75 | 160.69 | 0.026 |
| Dark illuminance–Normal illuminance | -0.53 | 0.75 | 160.69 | < .0001 |
| European style decoration | | | | |
| Bright illuminance–Dark illuminance | 0.53 | 0.75 | 160.69 | 0.755 |
| Bright illuminance–Normal illuminance | -2.18 | 0.75 | 160.69 | 0.011 |
| Dark illuminance–Normal illuminance | -2.72 | 0.75 | 160.69 | 0.001 |
| Bright light illuminance | | | | |
| Chinese style—European style | 3.72 | 0.76 | 112.54 | < .0001 |
| Normal light illuminance | | | | |
| Chinese style—European style | -0.43 | 0.76 | 112.54 | 0.0005 |
| Dim light illuminance | | | | |
| Chinese style—European style | -2.74 | 0.76 | 112.54 | 0.5743 |

**Table 5. ANOVA results for comfort under different types of stimulation.**

| Variables | Df | MS | F | p |
|---|---|---|---|---|
| 2×3×2×2 ANOVA results for comfort score | | | | |
| Light illuminance | 2 | 883.68 | 717.61 | < .0001*** |
| **Variables** | **Difference** | **SE** | **df** | **p** |
| Light intensity after contrast test | | | | |
| Bright illuminance–Dark illuminance | 2.75 | 0.15 | 84 | < .0001 |
| Bright illuminance–Normal illuminance | -0.06 | 0.15 | 84 | 0.93 |
| Dark illuminance–Normal illuminance | -2.81 | 0.15 | 84 | < .0001 |
| **Variables** | **Df** | **MS** | **F** | **p** |
| Wall color | 1 | 22.00 | 11.87 | < .0001*** |
| **Variables** | **Difference** | **SE** | **df** | **p** |
| Post-event comparison test of wall color | | | | |
| Cool color—Warm color | -0.29 | 0.05 | 42 | < .0001 |
| **Variables** | **Df** | **MS** | **F** | **p** |
| Decoration style | 1 | 28.21 | 22.91 | < .0001*** |
| **Variables** | **Difference** | **SE** | **df** | **p** |
| Post-event comparison test of decoration style | | | | |
| Chinese style—European style | 0.33 | 0.09 | 42 | 0.0009 |
| **Variables** | **Df** | **MS** | **F** | **p** |
| Music genre | 1 | 2.94 | 2.39 | .122 |

Note.

***p< 0.001

**p< 0.01

*p< 0.05.df is the degrees of freedom; MSE is mean square error; The F value is the statistic for the F test; Difference is the difference in the post hoc comparison test; SE is Standard Error.

**Table 6. Interaction effects and post hoc ANOVA results.**

| Variables | Df | MS | F | p |
|---|---|---|---|---|
| 2×3×2×2 ANOVA results for comfort score | | | | |
| Light illuminance: Wall color | 2 | 17.68 | 14.36 | $< .0001^{***}$ |
| **Variables** | **Difference** | **SE** | **df** | **p** |
| Post-mortem contrast test of lighting intensity and wall color | | | | |
| Bright light illuminance | | | | |
| Cool color- Warm color | -0.77 | 0.08 | 126 | $< .0001$ |
| Normal light illuminance | | | | |
| Cool color- Warm color | -0.23 | 0.08 | 126 | 0.0061 |
| Dim light illuminance | | | | |
| Cool color- Warm color | 0.13 | 0.08 | 126 | 0.1277 |
| **Variables** | **Df** | **MS** | **F** | **p** |
| Light illuminance: Music genre | 2 | 0.33 | 0.27 | .767 |
| Wall color: Music genre | 1 | 0.11 | 0.09 | .763 |
| Decoration style: Music genre | 1 | 0.08 | 0.07 | .797 |
| Light illuminance: Wall color: Decoration style | 2 | 13.20 | 10.72 | $< .0001^{***}$ |
| Light illuminance: Wall color: Music genre | 2 | 0.02 | 0.02 | .980 |
| Light illuminance: Decoration style: Music genre | 2 | 0.30 | 0.25 | 0.781 |
| Light illuminance: Wall color: Decoration style:music genre | 2 | 0.35 | 0.28 | .755 |

Note.

$^{***}$p$< 0.001$

$^{**}$p$< 0.01$

$^{*}$p$< 0.05$.df is the degrees of freedom;MSE is mean square error; The F value is the statistic for the F test; Difference is the difference in the post hoc comparison test; SE is Standard Error.

higher comfort than cool colors ($\bar{x}_{\text{bright illuminance}-\text{cool color}}$ = 4.122), with no significant difference between Chinese or European styles. However, under the influence of dim lighting, subjects' preferences for both color and style were not present (see Tables 6–8).

Considering the interaction of lighting, wall color and decoration style, according to the interaction chart of the three, it could be seen that it was best to choose bright lighting and cool colored walls for a Chinese style hotel. European style hotels were more suitable for normal illumination under warm color (see Fig 3).

## Discussion

This study offers several potential practical and theoretical contributions. Additional evidence is proposed to suggest that hotel consumers may be influenced by color, lighting, and decoration style. Combining the results of the eye-movement data and questionnaire data analysis, both proved that lamp illumination and wall colors affected the subjects' visual comfort, the difference between the two findings was that the questionnaire data also confirmed that the decoration style significantly affected the subjects' comfort. Therefore, hypotheses H1 and H3 were supported. Meanwhile, Elliot et al. (2012) argued that people's reactions to color were always unconscious, and people could not control their emotional and physiological reactions to color [42]. Knez et al. (2002) believed that indoor lighting could affect mood and physiological function, which was consistent with the results of our eye movement experiments, and verified the scientific rationality of introducing the measurement method of physiological indicators [25].

**Table 7. Interaction and post-hoc ANOVA results of lighting illuminance and decoration style.**

| Variables | Df | MS | F | p |
|---|---|---|---|---|
| 2×3×2×2 ANOVA results for comfort score | | | | |
| Light illuminance: Decoration style | 2 | 5.04 | 4.09 | .017* |
| **Variables** | **Difference** | **SE** | **df** | **p** |
| Post-mortem contrast test of lighting intensity and decoration style | | | | |
| Chinese decoration style | | | | |
| Bright illuminance–Dark illuminance | 2.96 | 0.17 | 119.58 | < .0001 |
| Bright illuminance–Normal illuminance | 0.16 | 0.17 | 119.58 | 0.6260 |
| Dark illuminance–Normal illuminance | -2.80 | 0.17 | 119.58 | < .0001 |
| European style decoration | | | | |
| Bright illuminance–Dark illuminance | 2.55 | 0.17 | 119.58 | 0.0006 |
| Bright illuminance–Normal illuminance | -0.27 | 0.17 | 119.58 | 0.2605 |
| Dark illuminance–Normal illuminance | -2.81 | 0.17 | 119.58 | < .0001 |
| Bright light illuminance | | | | |
| Chinese style—European style | 0.61 | 0.12 | 103.76 | < .0001 |
| Normal light illuminance | | | | |
| Chinese style—European style | 0.19 | 0.12 | 103.76 | 0.1393 |
| Dim light illuminance | | | | |
| Chinese style—European style | 0.20 | 0.12 | 103.76 | 0.1165 |

**Table 8. Interaction and post hoc ANOVA results of wall color and decoration style.**

| Variables | Df | MS | F | p |
|---|---|---|---|---|
| 2×3×2×2 ANOVA results for comfort score | | | | |
| Wall color: Decoration style | 1 | 5.46 | 4.44 | .035* |
| **Variables** | **Difference** | **SE** | **df** | **p** |
| Post-mortem contrast test of wall color and decoration style | | | | |
| Chinese decoration style | | | | |
| Cool color- Warm color | -0.15 | 0.07 | 82.75 | 0.0453 |
| European style decoration | | | | |
| Cool color- Warm color | -0.44 | 0.07 | 82.75 | < .0001 |

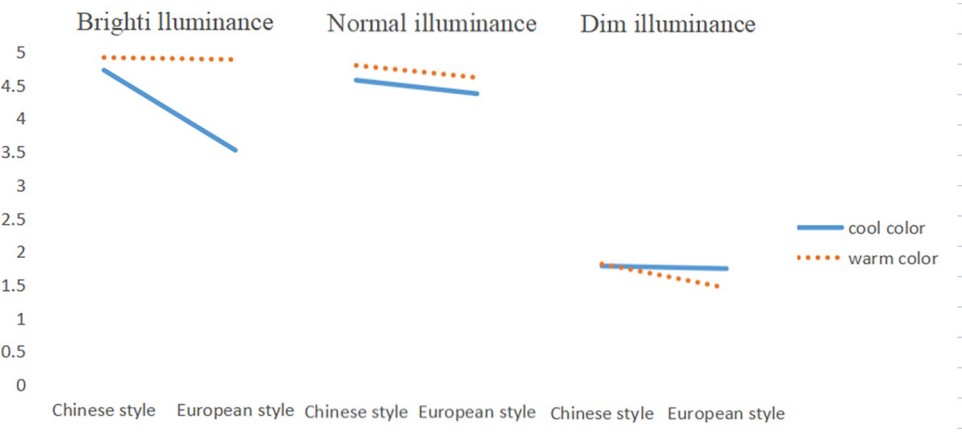

**Fig 3. Three-factor comfort interaction diagram.**

The results also confirm an interesting phenomenon when light illuminance, wall color and decoration style interact. Lighting illumination and wall tones should be adjusted according to the style of the hotel. When the illuminance of the lamp interacted with the wall hue, there was no significant difference in the preference for the warm and cool hues under dim illuminance, which was consistent with the findings of Babin et al. (2003). There was an interaction between light and color, the light would weaken the color effect, and any negative effects of color and light could be overcome by introducing other factors that could enhance evaluation and excitement [40]. Thus, the hypothesis H2a and H2b were supported.

Demoulin (2011) believed that music consistent with the service environment could create a more comfortable atmosphere [43]. In contrast, our findings demonstrated there was no bidirectional relationship between decorative style and musical style. There was no significant difference between Chinese or European music played in European style hotels. Thus, the hypothesis H4a, H4b and H4c were not supported.

The main reasons for the different conclusions reached by the eye movement index and the subjective questionnaire are as follows: according to the "emotion- cognitive-motor response" model proposed by Izard (2013), emotion is the basic motivation, and cognition plays an important role. The integration of emotional, cognitive, and motor systems produces certain experiences, emotions, and responses [23]. Compared with the illuminance of lamps and the color of walls, personal perception of decoration style is acquired through acquired learning, and there is a more complex cognitive judgment process. Among them, higher cognitive functions such as perception and memory retrieval play a more important role in the perception process. Therefore, the differences in participants' responses to different decoration styles are more influenced by their previous experiences and preferences. This difference is not obvious in the eye movement index, but it is more obvious in the subjective judgment. This also proves the necessity of combining subjective questionnaires with objective physiological indicators.

This study employs a combination of subjective questionnaire and physiological indicators to measure visual comfort. Compared with the previous method of utilizing subjective questionnaire only, the visual comfort is quantified more scientifically. The experimental results show that there are indeed differences in the measurement results of subjective questionnaires and physiological indicators. Therefore, from the perspective of emotion, this paper introduces visual comfort, uses the combination of subjective and objective methods to measure emotions, it makes up for the shortcomings of the existing subjective cognitive perspective, and further provides evidence that the illuminance of the lamp, the color of the wall and the style of decoration will affect the visual comfort of consumers.

By designing different lamp illuminance, wall color and music to obtain the consumer's unexpected experience, it has the advantages of low cost and easy operation. Hotel managers should consider overall factors during designing the lobby and try to use cool or warm colors, bright lights and soft music to create a warm atmosphere according to the style of the hotel.

This study also has some limitations. Although we demonstrated that cool and warm hues, lamp illumination, and decoration style could affect the visual comfort of consumers, in real life, color is not just hues, and many other factors will also affect the overall visual perception. Therefore, our future research will explore the effect of more other factors, such as the image of hotel staff, smell, light color, color brightness and saturation and so on. In addition, hotel rooms and restaurants are also important areas, the influencing factors of which will in our further study as well.

## Supporting information

**S1 Appendix. Hotel lobby models.**
(DOCX)

**S2 Appendix. Hotel visual comfort questionnaire.**
(DOC)

**S1 Data.**
(CSV)

## Acknowledgments

We would like to thank all the participants, editors and experts who provided valuable suggestions that improved the final version of this manuscript.

## Author Contributions

**Conceptualization:** Benhai Guo, Hongjuan Yin.

**Data curation:** Ziwen Geng.

**Formal analysis:** Ziwen Geng.

**Investigation:** Ziwen Geng.

**Methodology:** Ziwen Geng.

**Project administration:** Ziwen Geng.

**Software:** Ziwen Geng.

**Supervision:** Wei Le.

**Validation:** Ziwen Geng.

**Visualization:** Ziwen Geng.

**Writing – original draft:** Ziwen Geng.

**Writing – review & editing:** Ziwen Geng.

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
