## [Decision Letter · Decision Letter 0]

25 Jul 2022

PONE-D-22-06326Analysis of Factors Affecting Visual Comfort in Hotel Lobby from the Perspective of EmotionPLOS ONE

Dear Dr. GENG,

Thank you for submitting your manuscript to PLOS ONE. After careful consideration, we feel that it has merit but does not fully meet PLOS ONE’s publication criteria as it currently stands. Therefore, we invite you to submit a revised version of the manuscript that addresses the points raised during the review process.

I would like to sincerely apologise for the delay you have incurred with your submission. It has been exceptionally difficult to secure reviewers to evaluate your study. We have now received two completed reviews; the comments are available below. The reviewers have raised significant scientific concerns about the study that need to be addressed in a revision.

Please revise the manuscript to address all the reviewer's comments in a point-by-point response in order to ensure it is meeting the journal's publication criteria. Please note that the revised manuscript will need to undergo further review, we thus cannot at this point anticipate the outcome of the evaluation process.

We look forward to receiving your revised manuscript.

Kind regards,

Miquel Vall-llosera Camps

Senior Editor

PLOS ONE

Journal Requirements:

4. PLOS requires an ORCID iD for the corresponding author in Editorial Manager on papers submitted after December 6th, 2016. Please ensure that you have an ORCID iD and that it is validated in Editorial Manager. To do this, go to ‘Update my Information’ (in the upper left-hand corner of the main menu), and click on the Fetch/Validate link next to the ORCID field. This will take you to the ORCID site and allow you to create a new iD or authenticate a pre-existing iD in Editorial Manager. Please see the following video for instructions on linking an ORCID iD to your Editorial Manager account: " ext-link-type="uri" xlink:type="simple">https://www.youtube.com/watch?v=_xcclfuvtxQ"

Additional Editor Comments:

Some of the language used in the abstract of this paper might be overreaching and not supported by the data presented in study. Specifically, generalized statements such as 'the Chinese style room has always been more comfortable than the European style' without the proper context appear in the manuscript. Please revise to ensure that all statements in the manuscript are supported by the presented data (https://journals.plos.org/plosone/s/criteria-for-publication#loc-4). .

Reviewers' comments:

Reviewer's Responses to Questions

**Comments to the Author**

1. Is the manuscript technically sound, and do the data support the conclusions?

Reviewer #1: Partly

Reviewer #2: Partly

2. Has the statistical analysis been performed appropriately and rigorously? 

Reviewer #1: No

Reviewer #2: No

3. Have the authors made all data underlying the findings in their manuscript fully available?

Reviewer #1: Yes

Reviewer #2: Yes

4. Is the manuscript presented in an intelligible fashion and written in standard English?

Reviewer #1: No

Reviewer #2: Yes

5. Review Comments to the Author

Reviewer #1: Comments：

The study aims to investigate how the light illumination, wall tone, decoration style and music style

interactionally influenced the visual comfort by using a subjective measurement and specifically

eye tracking technology. It is a interesting meaningful topi in practical consumer behavior

management. However, I also saw major issues regarding the rational of the study, methodology,

and data analysis. My comments and questions that I present below:

Title:

1 he title needs to be improved as the perspective of emotion was not well reflected in the study.

Introduction

1: what’s the main rational for integrating the visual elements with auditory stimulus when to

measure visual experience? As the author mentioned there were many factors influencing the visual

comfort the author proposed the visual element that function as a more important role, such as

illuminance, color and decoration style. The auditory stimulus “music” was also involved, how does

it influence visual comfort?

2 What the definition of “Visual comfort” ? if people stand in a hotel lobby, their feelings would be

affected with visual elements but the visual comfort was not the whole story. I guess the author aims

to measure the psychological and physiological comfort. However, the approach employed was

mainly influence the visual experiences as only the pictures but not the real physical stimuli were

exposed. In other word, it would be flexible to test these effects by using the virtual reality display

technology and the current study paradigm largely influence visual comfort and perception.

3. For each visual elements, in addition to light level, the spectrum such as warm light or cool light

would also be a factor influencing visual experience, the author also stated it in the introduction, but

it was not involved in manipulation of the light of the model.

4. What the decoration style was employed for Chinese and European style？The decoration style

itself was indeed created with including multiple visual elements, such as light level, color and

furniture etc. that would also moderate the visual experience, how did the author deal with that when

to manipulate the experimental conditions?

5.Generally, the literature review on the effect of visual elements on comfort and mood need to be

improved, and what the effects already have been investigated and what need to be explored?

Especially, how these factors interactionally influenced visual experiences that the authors

hypothesized more subjectively.

6.The main measurement of the current study was the visual comfort. The author stated a lot on

measurement of mood, please rephased it to be more concise and also I missing how the main visual

elements influence mood in previous literature review.

7. The hypothesis could be given at the end of the introduction to make the literature review more

continuously.

8. The reference style presented in the introduction as well as elsewhere need to be improved by

reference management tool with the requirement of the journal.

Method

1.How was the sample size decided? Did you employed a G-power analysis to see how many

participants were required with the current design?

2. The participants had hotel experiences but did they had preferences to decoration style that might

be a response bias in evaluating the experimental stimulus.

3.What’s the study design? Within or between-subjects design and how the pictures were arranged

for each participant? Was it randomized or presented with a fixed order?

4. The experiment was approved by the relevant department, what’s the department? Was there a

ethical statement as human participants were involved?

5. what the 3D pictures look like? Did them matched with brightness, familiarity etc that would be

potential factors influencing visual experiences and affective state? Also, was there only one sample

for each kind of picture with manipulating illuminance, color and decretory style?

6. The hall model employed in the pre-experiment were same with that in the current experiment?

What the aim of the pre-experiment? To verify the questionnaires? Did participants assessed the

illuminance, color and decretory style in the pre-experiment? What the differences or adjustment

had been done for formal experiment? Please state that in the method section.

7.Please rephase the statement of the procedure as what had been done before, during and after the

experiment? Also the five set of experiment seems to be five session with each session including

two block. It’s confusing that the experiment consisted five set of experiment.

6. What’s the 3D picture looks like? Please give samples, probably in the supplementary file.

7. The full name of the PANAS needs to be give where it first occurred.

8. How participants assessed their comfort? By using standard questionnaire and what the response?

9. What the interested indicators of the eye movement recording and what did they mean? As the

reflection of visual comfort or mood？please state it in the method as well as in the introduction.

10. What the statistical method and tool employed for data analysis was missing?

Results

1.The structure of the results section for eye movement indicators as well as for the subjective

indicators needs to be adjusted. As for main factors except for illuminance, only two levels were

involved, the post-hoc contrasts did not need. Please added the relationship after the corresponding

main effect presented. For interaction, please added the statement of the post-hoc contrast following

the corresponding interaction effect. It would be clearer to separate main effects and interaction

effects into two paragraphs. If the post-contrast were employed, the Bonferroni correction need to

be applied.

2.What was the analysis method? LMM model or ANOVA? Was the df presented in the table?

3.Please correct the front style of statistical symbols, P should be p and all symbols need to be in

italic. If the exact p values were given, the significant level did need any more.

4. Also the statistics were presented in the main text, the value resented in the table was redundant.

It would be better to added the descriptive for each indicator in the table.

5. please remove the statements regarding the hypothesisin the results section, it would be redundant

and can be added in the discussion

Discussion

1. The discussion needs to be rephased and improved as what the main finding of the current study?

2. What the newly effects were revealed in the current study, were it the same or different from

what the finding reported in previous studies and why?

3. Objective eye movement indicators and subjective assessment of visual comfort and mood were

employed, however, I did miss the

4. The author explained the effects of decoration style with the prior experiences of participants?

As I mention in previous comments, were the experience not pre-assessed or added it as

covariate? Please also added it as the limitation.

5. The current study showed much significance in practical sensory, market of hotel as well as in

theoretical work. Please state it more concise.

In addition, the writing expression need to be improved throughout the draft, please ask one native

speaker of English to polish the writing before re-submit.

Reviewer #2: In the research, the emotional effects of many factors such as wall color, light brightness, light tone (warm-cold), music in the hotel lobby were tried to be measured. In the research, scale questions to measure emotions and eye tracking as a technical measurement tool were used.

The research has a very valuable design and content in terms of contributing to both the scientific field and the hotel industry.

However, some of the following issues have been identified;

1. Experimental studies, color, light, music, etc. Studying many factors together affects the validity of the experiment. Therefore, there is a problem with the limitations of the research.

In the second study, although the sample size seems to be sufficient in an experimental study, the design of the experiment as a control group and a subject group is not clear. What kind of experiment did the author/s do? The explanation of the method proving the validity of the experiment could not be understood.

3. Brightness and warm-cold tones are emphasized in the use of colors and lights. However, colors must be expressed by giving technical scale ranges. Expressing it as hot-cold is insufficient. Regarding the light, the degree of brightness should also be shown by giving the value of the lamps used.

4. Although the combination of many factors causes the complexity of the boundaries of the research, if the above deficiencies are completed in the research, an understandable situation will occur.

5. The sources used by the authors are quite old. In particular, there are many specific journals in this field such as ColorResearch. And in these journals, there are color and light studies with high scientific value. The bibliography can be enriched.

I hope my comments do not discourage the authors, I wish you good luck.

6. PLOS authors have the option to publish the peer review history of their article (what does this mean?). If published, this will include your full peer review and any attached files.

Reviewer #1: **Yes: **Taotao Ru

Reviewer #2: No

---

## [Author Response · Author response to Decision Letter 0]

8 Sep 2022

We have revised the manuscript and responded to all the reviewers' comments point by point.

---

## [Decision Letter · Decision Letter 1]

28 Sep 2022

PONE-D-22-06326R1

Analysis of Factors Affecting Visual Comfort in Hotel Lobby

PLOS ONE

Dear Dr. GENG,

We have now completed the review process for your manuscript. Based on these comments, we have determined that we cannot accept the paper in its current form. However, we believe that the study has some inherent value and may be publishable if revised appropriately.

If you are willing to undertake the recommended revisions, I would be pleased to reconsider the manuscript for possible publication. If you disagree with any of the recommendations, I would also be willing to consider a rebuttal to any of the points made. For your guidance, the comments obtained during peer review are appended below. We hope that you find these helpful

It is important to note that we cannot make any promises as to whether a revised manuscript will be accepted and you should consider the extent to which you can address the comments below before revising your work for resubmission.

If you decide to revise the work, please submit a detailed list of changes for each point raised when you submit the revised manuscript. Please also highlight where the text has been changed in the resubmitted article - this will help to streamline the peer review process and minimize any delays. The points raised during the review process.

Although the author addressed all my comments, the main issues 1) The reational of the study combining multiple factors together in a study as each of the factor had influence on visual comfort; 2) The 3D pictures were employed to exposure different visual elements to participants, however the presented pictures seems not definitely reflect the manipulation of the factors such as color, and also several pictures with such higher brightness or darknee that they were difficult to see clear, all these would limite the possible experiences of participants; 3) The 2*2*2*2*3 within subjects design was employed leading a quiet complexy analysis and the interaction effects would be meaningless; 4) The sample size was relatively small; 5) The analysis need to be improved and the explanation also need to be expand. In addition, the The readability of the most sentences still need to be polish.

If applicable, we recommend that you deposit your laboratory protocols in protocols.io to enhance the reproducibility of your results. Protocols.io assigns your protocol its own identifier (DOI) so that it can be cited independently in the future. For instructions see: https://journals.plos.org/plosone/s/submission-guidelines#loc-laboratory-protocols. Additionally, PLOS ONE offers an option for publishing peer-reviewed Lab Protocol articles, which describe protocols hosted on protocols.io. Read more information on sharing protocols at https://plos.org/protocols?utm_medium=editorial-emailutm_source=authorlettersutm_campaign=protocols.

We look forward to receiving your revised manuscript.

Kind regards,

Thiago Fernandes, MD, Sp. Neur, PhD

Academic Editor

PLOS ONE

Additional Editor Comments:

Thank you for your submission. Please respond to all Reviewers’ comments AND highlight them. A few things should be emphasised:

1. Enlist a native English speaker to edit grammar throughout the text;

2. I’d highly recommend you to reframe your analyses. That’s still not clear

3. You might want to use Bayesian. The use of multivariate or other Bayesian approach on JASP is simple and it’s a ready-and-quick-to-use software

4: I’d suggest you to carefully address the remaining concerns from the Reviewer

Reviewers' comments:

Reviewer's Responses to Questions

**Comments to the Author**

1. If the authors have adequately addressed your comments raised in a previous round of review and you feel that this manuscript is now acceptable for publication, you may indicate that here to bypass the “Comments to the Author” section, enter your conflict of interest statement in the “Confidential to Editor” section, and submit your "Accept" recommendation.

Reviewer #1: All comments have been addressed

Reviewer #2: All comments have been addressed

2. Is the manuscript technically sound, and do the data support the conclusions?

Reviewer #1: Partly

Reviewer #2: Yes

3. Has the statistical analysis been performed appropriately and rigorously? 

Reviewer #1: No

Reviewer #2: Yes

4. Have the authors made all data underlying the findings in their manuscript fully available?

Reviewer #1: Yes

Reviewer #2: Yes

5. Is the manuscript presented in an intelligible fashion and written in standard English?

Reviewer #1: No

Reviewer #2: Yes

6. Review Comments to the Author

Reviewer #1: (No Response)

Reviewer #2: Dear Author/s,

Thank you for your detailed revisions by considering the suggestions presented to you in the initial evaluation.

Data on light and colors are given in detail. Necessary revisions were made in the method section. Some subjects were expressed within the limitations of the research. The literature has been enriched within the framework of related topics.

I recommend that you review it once again for typos.

I wish you good luck.

7. PLOS authors have the option to publish the peer review history of their article (what does this mean?). If published, this will include your full peer review and any attached files.

Reviewer #1: **Yes: **Taotao Ru

Reviewer #2: **Yes: **Assoc. Prof. Dr. Bilsen Bilgili, Kocaeli University/Turkey

---

## [Author Response · Author response to Decision Letter 1]

8 Nov 2022

Response to Reviewer #1: 

Thank you again very much for the insightful comments on our paper. We have revised our paper carefully according to these suggestions. In this note, we will first list your comments and then outline what we have done in response to these comments.

1)The reational of the study combining multiple factors together in a study as each of the factor had influence on visual comfort.

Response: Thank you very much for your comment. The effects of lighting on mood and comfort, the effects of color on mood and comfort, and the effects of music on mood and comfort are all well-researched.

(1)Color, emotion and visual comfort in service setting

Color is one of the key visual dimensions of the environment that has an impact on behavior and emotions. The emotional effect of color is particularly important in the hospitality industry because emotions comprise a powerful affective component of customer satisfaction. The differences between two contrast hues( blue and red) with two levels for saturation and brightness were examined in an experimental study. The results indicated participants enjoyed staying in blue hotel rooms more than in the red ones. The blue room was more pleasant, comfortable, and relaxing(Katsiaryna Siamionava et al., 2018). In the hotelscape, color had been shown to significantly impact customers’ aesthetic perceptions (Alfakhri et al., 2018). A comparison of warm and cool colors revealed that cool colors were not aesthetically superior to warm colors in the luxury hotel context and warmer colors were more pleasing(Warakul Tantanatewin and Vorapat Inkarojrit, 2018; Dongyoup Kim et al., 2020). However, an academic study also demonstrated that no statistically significant effects of color and temperature on comfort results within the tested settings(Arianna Latini et al., 2021).

(2)Light illuminance, emotion and visual comfort in service setting

Lighting was an indispensable factor in the design of the hotel lobby. Lighting affected the overall impression and perceived comfort of the lobby(Akita, 2019; Countryman and Jang, 2006). The level of visual comfort in lighting in a residential context turned to be more influenced by the color temperature and illuminance compared to other factors(Yoon et al., 2014). Under high illumination, the level of anxiety would decrease, which was more conducive to relieving fatigue. Providing a user-friendly and user-controlled lighting system for hotel rooms helped improve customer satisfaction(Akita, 2019). Light not only affected people’s emotions, but the interaction between light and sound also affected people’s collection of target information. Low-illumination light was more likely to cause visual fatigue. Lower contrast colors with darker lighting and simple decoration made it easier to create a relaxed and romantic atmosphere. Dim light with soothing music could create a relaxed and comfortable atmosphere(Siamionava et al., 2018). A lower color temperature would create a higher sense of security, positive feelings and restfulness(Akita, 2019). While some scholars found that brighter lights were more conducive to positive emotions( Figueiro, 2016). Different environments have different lighting requirements. 

(3)Color, light illuminance, music, emotion and visual comfort in service setting

There were interactions between the senses, the interaction between senses was mainly reflected in their direct impact on emotions( Kantonoet et al., 2019). The tone stimulus could alter the pleasantness rating of stimuli from other senses(Dematte et al., 2007; Pollatos et al., 2007; Logeswaran and Bhattacharya, 2009). For example, congruent sounds could enhance odor pleasantness to a higher degree than can incongruent sounds(Seo, 2011). Music generally interacted with other environmental factors and music emotion was significantly correlated with lighting(Morin et al., 2007; Hsiao et al., 2017). There were also structural correspondences between color and music(such as emotional expression, hierarchical organization and contrast)(Sebba,1991). Demoulin(2011) studied the consistency of music and the overall atmosphere, and the results showed that when coordinated with the service environment and music, the relaxing, calm and pleasant atmosphere would be created(Demoulin, 2011).

Although music is not a visual factor, it will have an impact on the visual comfort of lighting, color and style. What we see in real life is the overall comfort of the hotel lobby, not the comfort brought by individual lights or colors. Therefore, these four factors are inseparable, and we put them in a model to study.The results of the study by Barry J. Babin et al. (2003) showed that the results change considerably when the effects of lighting in combination with color was taken into account. The experimental results proved that there was indeed an interaction between lamp illumination, wall tone and decoration style(Babin et al., 2003).

2)The 3D pictures were employed to exposure different visual elements to participants, however the presented pictures seems not definitely reflect the manipulation of the factors such as color, and also several pictures with such higher brightness or darknee that they were difficult to see clear, all these would limite the possible experiences of participants.

Response: Thank you very much for your comment. A 3D hotel lobby model was based on a compilation of typical hotel lobby. The lobby consisted of the basic attributes of a typical hotel lobby: waiters, decorative lights, and front desk cabinets. Therefore, the models did not definitely reflected the manipulation of color and other factors, but the model tried to keep other factors consistent, except for the light intensity, wall tone and decoration style.

The specific intensity of light illumination was determined according to the opinions of pre experiment experts and subjects. The reason why some pictures had high or low brightness was that there were three levels of light illumination: bright illumination, normal illumination and dim illumination. The experimental results showed that the illumination comfort of bright light was relatively high, while that of dim light was the lowest.

3)The 2*2*2*2*3 within subjects design was employed leading a quiet complexy analysis and the interaction effects would be meaningless.

Response: Thank you very much for your comment. In reality, consumer comfort is determined by the environment after the interaction of multiple factors, rather than a single factor. Lighting, color and decoration style are are inseparable in our visual perception. Multisensory interaction studies and multifactor interaction studies are increasing. We refer to multifactor interaction studies from leading journals in various fields. For example, in the field of sensory marketing, there were many papers that studied the repetitive experiments within subjects with multiple factors and studied the impact of interaction, such as Wonyoung Yanga and Hyeun Jun Moon(2019) investigated the influence of multisensory interaction on acoustic comfort, thermal comfort, visual comfort, and indoor environmental comfort. Their experimental design was 3(Homogenous room temperatures: 20, 25, and 30 °C )×3(Illuminance levels: 150,500, and 1000 lx) × 4( Different types of sound : babble, fan, music, and water) × 4(Sound levels: 45, 55, 65, and 75 dBA) (Yang W and Moon HJ, 2019).

Katsialyna Siamonava et al. (2018) studied the impact of space color on guests’ perception of hotel rooms, and they adopted 2c×2×2 design . The results showed that the three interactions were not significant, but at the 1% confidence level, two of the two interactions were significant(Katsialyna Siamonava et al., 2018).

Holger Roschk et al.(2017) examined the effects of music(presence vs. absence), scent (presence vs. absence), and colors (warm: red, orange, and yellow vs. cool: green, blue, violet,and white) on shopping outcomes. The results showed that being in an environment with music or scents produced higher ratings of pleasure, satisfaction, and behavioral intention compared to environments without these conditions(Holger Roschk et al., 2017).

The study design by Barry J. Babin et al.(2003) was 2(color: blue, orange) × 2(light: bright, soft) ×2 (price: high price, low price). The results of the ANOVA and post hoc tests indicated an interaction between the three, the results changed substantially when the effect of lighting in combination with color was considered. The use of soft lights with an orange interior generally nullified the ill effects of orange and produced the highest level of perceived price fairness while controlling for price(Babin et al., 2003).

4)The sample size was relatively small.

Response: Thank you very much for your comment. The sample size of 43 people is indeed relatively small, but it also meets the requirement of the number of people in the consumer eye movement experiment. The proof that we have the required number of subjects is as follows, for example, Peter Walla et al.(2011) used an eye-tracking experiment to study emotional and physiological responses to brand attitudes in 29 subjects (Peter Walla et al., 2011). Qiuzhen Wang et al. (2014) studied eye-tracking of website complexity from a cognitive load perspective, with 42 college students as subjects(Qiuzhen Wang et al., 2014). Borisuit et al.(2014) showed that workplace lighting conditions affected various factors related to job satisfaction, productivity, and well-being. Lighting conditions affected visual comfort, mood, alertness, and well-being, 25 People were tested(Borisuitet et al. 2014). Research by Vytautas Abromavičius and Artūras Serackis(2018) showed that pupil diameter and the number of fixation points can be used to measure visual comfort. The “uncomfortable” visual comfort score had the highest pupil size and lower fixation points. There were 24 subjects in the eye movement experiment(Vytautas Abromavičius and Artūras Serackis, 2018). Cajochen et al.(2019) studied that lighting conditions affected subjects’ visual comfort, mood, and sleep intensity. The number of subjects was 15(Cajochen, 2019). Yu Wang et al.(2020) investigated the interactive effects of illumination and the associated color temperature on observers’ color preference and degree of white light perception with 30 subjects(Yu Wang et al., 2020). Arianna Latini et al.(2021) used immersive virtual reality to assess productivity and comfort in the workplace with 23 subjects(Arianna Latini et al., 2021).

5)The analysis need to be improved and the explanation also need to be expand. In addition, the The readability of the most sentences still need to be polish.

Response: Thank you very much for your advice. We have added new interaction charts and refined the analysis and interpretation of the data. Babin et al. (2003) applied ANOVA and post hoc tests to verify consumer responses to various combinations of color, lighting, and price point(Babin et al., 2003). Han-Seok Seo et al.(2011) used ANOVA and post hoc tests to verify the integration effect of auditory and olfactory sensations(Han-Seok Seo et al., 2011). Katsiaryna Siamionava et al.(2018) used ANOVA to study the effect of colors of different hue, saturation and brightness on consumers(Katsiaryna Siamionava et al., 2018). 

Our data analysis method was to apply the aov function in the R language to do repeated ANOVA and post hoc tests.

We sent this article to another native English speaker for proofreading, and revised it according to professional suggestions.

Response to Reviewer #2: 

Thank you again very much for the insightful comment on our paper. We sent this article to another native English speaker to proofread and fix the typos and other issues.

Response to Editor: 

1.Enlist a native English speaker to edit grammar throughout the text;

Response: Thank you very much for your advice. 

We sent this article to another native English speaker for proofreading, and revised it according to professional suggestions.

2. I’d highly recommend you to reframe your analyses. That’s still not clear;

3. You might want to use Bayesian. The use of multivariate or other Bayesian approach on JASP is simple and it’s a ready-and-quick-to-use software;

4: I’d suggest you to carefully address the remaining concerns from the Reviewer.

Response: Thank you very much for your advices. 

We have added new interaction charts and refined the analysis and interpretation of the data. Our data analysis method was to apply the aov function in the R language to do repeated ANOVA and post hoc tests.

The theoretical basis for our application of ANOVA is as follows, Babin et al. (2003) applied ANOVA and post hoc tests to verify consumer responses to various combinations of color, lighting, and price point(Babin et al., 2003). Han-Seok Seo et al.(2011) used ANOVA and post hoc tests to verify the integration effect of auditory and olfactory sensations(Han-Seok Seo et al., 2011). Katsiaryna Siamionava et al.(2018) used ANOVA to study the effect of colors of different hue, saturation and brightness on consumers(Katsiaryna Siamionava et al., 2018). Wonyoung Yang and Hyeun Jun Moon(2019) used ANOVA and post hoc tests to verify the combined effects of acoustic, thermal, and lighting conditions on discrete sensory comfort and the overall indoor environment(Yang and Hyeun Jun Moon, 2019).

We have submitted these refereed refereed journal papers as supplementary documents.

---

## [Editor Report · Decision Letter 2]

9 Nov 2022

PONE-D-22-06326R2Analysis of Factors Affecting Visual Comfort in Hotel LobbyPLOS ONE

Dear Dr. GENG,

Thank you for submitting your valuable work, and also for the careful edits.From my standpoint, all of the issues were properly eased. However, I still need to bring your attention to some issues. I can anticipate these are quick-to-solve, but essential to refine soundness of your study. I would be happy to look again after your edits so then we can proceed with your study. 1) Although I completely understand why you picked up the term - why don’t you use “visual setting” or “conformation”. Please clarify with a theoretical perspective, there is no need to emphasise past studies and the “common term”2) The authors mentioned about colour processing (and perception, per se) - so then I ask the authors to check references on confounding factors. For example, were excluded if had any substance use (references), use of meds (references) and so it goes. 3) I still think ANOVA isn’t correct. I understand that you pointed out some refs that run ANOVA, but you have more than one DV. Seems really different from a 2 x 2 … Please consider MANOVA, effect sizes and CIs4) Please place limitations and further directions for researchers and readers on a smoother way (for example, you have some limitations here and there, but others need to interpret properly and, if want to, replicate basing on your arguments)5) Check the references that are not in the text  Finally, to speed up the flow, you can send it again soon as you can.

We look forward to receiving your revised manuscript.

Kind regards,

Thiago Fernandes

PLOS ONE

Journal Requirements:

Additional Editor Comments (if provided):

Please read my comments.

---

## [Author Response · Author response to Decision Letter 2]

14 Dec 2022

Response to Reviewer : 

Thank you again very much for the insightful comments on our paper. We have revised our paper carefully according to these suggestions. In this note, we will first list your comments and then outline what we have done in response to these comments.

1)Although I completely understand why you picked up the term - why don’t you use “visual setting” or “conformation”. Please clarify with a theoretical perspective, there is no need to emphasise past studies and the “common term”.

Response: Thank you very much for your advice. There were many studies on “visual setting” or “conformation”, such as Katsiaryna Siamionava et al. (2018) showed that wall color in visual setting affected consumer perception. Dongyoup Kim et al. (2020) showed that the visual service setting in luxury hotels affected the behavior and emotions of the subjects. After summarizing the factors that affected hotel comfort, we found that visual setting affects consumer perception and played the most important role in sensory evaluation factors. However, visual comfort, which ultimately affected people’s motivation to consume and how visual setting affected visual comfort, was often overlooked. 

Light illumination, wall tones, and decorative style in the visual setting are the independent variables in this study. Our main objective is to investigate their effects on visual comfort and to explore the visual environment in which consumers’ visual comfort would be higher. We discuss the reasons for choosing visual comfort as the dependent variable in the paper, please check. Past relevant studies and definitions of visual comfort were the theoretical basis for our choice of eye movement metrics. If you feel that they are not needed for this study, we can remove them.

2)The authors mentioned about colour processing (and perception, per se) - so then I ask the authors to check references on confounding factors. For example, were excluded if had any substance use (references), use of meds (references) and so it goes.

Response: Thank you very much for your advice. Based on your suggestions, we have combed through the color-related literature de novo, categorized and summarized the existence of color interactions as well as non-interactions, and explained the reasons why we chose to study color-light interactions.

3)I still think ANOVA isn’t correct. I understand that you pointed out some refs that run ANOVA, but you have more than one DV. Seems really different from a 2 x 2 … Please consider MANOVA, effect sizes and CIs.

Response: Thank you very much for your advice. Based on your advice, we have re-analyzed the data using MANOVA, please check.

4)Please place limitations and further directions for researchers and readers on a smoother way (for example, you have some limitations here and there, but others need to interpret properly and, if want to, replicate basing on your arguments).

Response: Thank you very much for your advice. We have freshly combed this section of the article on limitations to provide further guidance for researchers and readers.

5) Check the references that are not in the text.

Response: Thank you very much for your comment.We have checked all references and updated some of them, please check them.

---

## [Editor Report · Decision Letter 3]

29 Dec 2022

Analysis of Factors Affecting Visual Comfort in Hotel Lobby

PONE-D-22-06326R3

Dear Dr. GENG,

We’re pleased to inform you that your manuscript has been judged scientifically suitable for publication and will be formally accepted for publication once it meets all outstanding technical requirements.

Kind regards,

Thiago P. Fernandes, PhD

Academic Editor

PLOS ONE

Additional Editor Comments (optional):

Thank you for addressing all concerns and refined this version. Please check references and grammar to speed up typesetting.
---

## [Editor Report · Acceptance letter]

6 Jan 2023

PONE-D-22-06326R3 

Analysis of Factors Affecting Visual Comfort in Hotel Lobby 

Dear Dr. Geng:

I'm pleased to inform you that your manuscript has been deemed suitable for publication in PLOS ONE. Congratulations! Your manuscript is now with our production department. 

Kind regards, 

on behalf of

Dr. Thiago P. Fernandes 

Academic Editor

PLOS ONE